# Interfacial Catalysis during Amylolytic Degradation of Starch Granules: Current Understanding and Kinetic Approaches

**DOI:** 10.3390/molecules28093799

**Published:** 2023-04-28

**Authors:** Yu Tian, Yu Wang, Yuyue Zhong, Marie Sofie Møller, Peter Westh, Birte Svensson, Andreas Blennow

**Affiliations:** 1Department of Plant and Environmental Sciences, University of Copenhagen, DK-1871 Frederiksberg C, Denmark; 2Enzyme and Protein Chemistry, Department of Biotechnology and Biomedicine, Technical University of Denmark, DK-2800 Kongens Lyngby, Denmark; 3Applied Molecular Enzyme Chemistry, Department of Biotechnology and Biomedicine, Technical University of Denmark, DK-2800 Kongens Lyngby, Denmark; 4Interfacial Enzymology, Department of Biotechnology and Biomedicine, Technical University of Denmark, DK-2800 Kongens Lyngby, Denmark

**Keywords:** starch, starch granules, amylase, enzyme kinetics, interfacial catalysis

## Abstract

Enzymatic hydrolysis of starch granules forms the fundamental basis of how nature degrades starch in plant cells, how starch is utilized as an energy resource in foods, and develops efficient, low-cost saccharification of starch, such as bioethanol and sweeteners. However, most investigations on starch hydrolysis have focused on its rates of degradation, either in its gelatinized or soluble state. These systems are inherently more well-defined, and kinetic parameters can be readily derived for different hydrolytic enzymes and starch molecular structures. Conversely, hydrolysis is notably slower for solid substrates, such as starch granules, and the kinetics are more complex. The main problems include that the surface of the substrate is multifaceted, its chemical and physical properties are ill-defined, and it also continuously changes as the hydrolysis proceeds. Hence, methods need to be developed for analyzing such heterogeneous catalytic systems. Most data on starch granule degradation are obtained on a long-term enzyme-action basis from which initial rates cannot be derived. In this review, we discuss these various aspects and future possibilities for developing experimental procedures to describe and understand interfacial enzyme hydrolysis of native starch granules more accurately.

## 1. Introduction

The staggering interest and developments in areas such as green chemistry, food digestion related to health impacts and recyclable materials have stimulated research into enzymatic hydrolysis of insoluble biopolymers, such as starch and lignocellulose. Enzymatic hydrolysis of native starch in a heterogeneous environment of solid interfaces is a reality in many biological and industrial processes, such as transitory starch degradation in plants [1], dietary digestion by animals [2], malting and fermentation processes [3], enzymatic modification for improving physical and functional characteristics [4], and low-temperature protocols for the production of glucose syrups or bioethanol [5].

Starch is commonly found in the form of granules ranging from 1 to 100 μm in size within leaves and non-photosynthetic tissues. The latter is typically considered storage starch, which is a vital source of nutrition for human consumption and is widely used in various industrial applications. The functional properties of storage starch vary significantly depending on its complex multi-level structure (Figure 1). Hydrolytic degradation of starch in the human gastrointestinal tract (GIT) occurs in consecutive oral, duodenal and small intestinal stages, leading to various maltooligosaccharides, maltose and glucose. However, the starch portion passing through the small intestine and entering the colon is defined as Resistant Starch (RS), providing an important energy source for the colonocytes and is beneficial for a healthy gut (Figure 2). Several enzymes are involved in the process of hydrolysis of starch to glucose in the GIT, such as salivary (HSA) and pancreatic (HPA) amylases hydrolyze starch in the oral and small intestine, respectively. These activities mainly provide maltooligosaccharides that are readily converted to glucose by the two brush border maltase-glucoamylase (MGAM) and sucrase-isomaltase (SI) heterodimeric enzyme complexes in the small intestine resulting in the uptake of glucose there [6]. RS is degraded by the complex ecosystem of microbes in the colon by producing starch-degrading enzymes, including α-amylase for α-1,4, type I pullulanase for α-1,6 and type II pullulanase for degradation of both α-1,4 and α-1,6 linkages [7]. High and controlled degradative resistance of the dietary fiber RS forms a general concept for its nutritional value. It is not clear, however, if the resistance of different types of RS stems from a slow turnover of the hydrolysis of glucosidic bonds, weak binding of enzymes, limited substrate accessibility for enzyme hydrolysis or a combination of these effects. Such knowledge is imperative for the development of new functional RS dietary fiber.

Enzyme kinetics provides the experimental link between structure and function in biocatalysis. Contrary to homogeneous catalysis, where substrate and enzyme are both in solution, the hydrolysis of insoluble solid substrates (e.g., starch granules) constitutes a heterogeneous (interfacial) catalytic process that is challenging because the molar concentration of the substrate cannot be defined unambiguously [10]. Classical Michaelis–Menten (M-M) kinetics typically measures catalysis in dilute solution, which is not characteristic of most natural situations. In particular, it is unclear whether conventional M-M theory can be applied, which requires a large excess of substrate. In this review, we discuss these various aspects and possible ways forward to build experimental systems to describe and understand interfacial enzyme hydrolysis of native starch granules more accurately. For example, as inspired by the hydrolysis of cellulose fibers by cellulases [11], we suggest a new kinetics approach to analyze granular starch degradation, which introduces an important factor that enumerates sites (in units of mol/g substrate) available for enzyme attack in the substrate suspension.

## 2. The Starch Granule

The starch granule is an exceptionally well-organized and compact polysaccharide energy source [12]. It forms the main component of most plant foods [13] and is thereby of tremendous importance for human wellbeing. The world annual production of raw starch reached a volume of 120.4 Mt in 2020 and is projected to reach 168.9 Mt by 2027 (https://www.reportlinker.com/p05485911/Global-Starch-Industry.html?utm_source=GNW, accessed on 20 December 2022).

The starch granule is a semi-crystalline multi-level entity with specific structural features, defined at the molecular level mainly by the essentially linear α-1,4 glucan amylose (AM) and the branched α-1,4;α-1,6 glucan amylopectin (AP) [14], at the 8–11 nm level by crystalline and amorphous lamellar and at the 0.1 μm scale by alternating amorphous and semi-crystalline growth rings. Finally, the size of the entire starch granule is 1–100 μm depending on the botanical origin [9] (Figure 1).

Models and representations of these different structural levels are continuously discussed (e.g., [15]). One important point is the currently debated description of the molecular structure of amylopectin as either the so-called cluster model [16,17,18] or as the more recent building block backbone model [9]. Both models agree that the double helices are oriented perpendicularly to the surface of the starch granules. While the cluster model entails a radial tree-like clustering organization of the branch chains in the amylopectin molecule, the backbone model suggests that long backbone chains are tangential to the direction of the double-helical structures of branch chains (Figure 1e). Accordingly, the long chains of the backbone structure form two-dimensional sheets on which non-clustered branched building blocks are attached from which shorter crystal-forming chain segments protrude in the perpendicular direction (in the building block backbone model, Figure 1), or the long chains have essentially the same orientation as the double helices and can penetrate several layers of double helices (in the cluster model). These segments are suggested to be randomly distributed from each other and have sufficient inter-branch space of less than nine glucose residues (degree of polymerization (DP) < 9 (5–8)) [9] to allow parallel double helices to be formed and crystallize. One such backbone/double-helix segment layer constitutes a 9-nm thick lamella, which creates a structural entity yielding concentric structures in the starch granule. The branching of amylopectin, the ratio and the chain length of amylose and amylopectin all play important roles in the granular architecture [9,14,19]. Amylose seems to be interspersed into the granular matrix, mainly in amorphous regions. However, little is known about how different molecular structures affect granular architecture and thelolytic susceptibility.

### 2.1. Nano-Level Structures

At the nano-level, the starch granule is assembled to contain mainly one of two different double-helical crystalline systems, namely the A- and B-type crystalline polymorphs, as determined by wide-angle X-ray scattering (WAXS) [19]. For storage starches, the A-type polymorph is typically found in cereal grain starches, while the B-type is found in tuber and root starches as well as in high-amylose (>50%) starches. Pulses (the edible seeds of plants in the legume family) may have a mixture of the two, a so-called C-type starch. High amylose starches also tend to have a third polymorph termed the V_h_-type polymorph, made of single helices, typically complexed with lipids [20]. As deduced from small angle X-ray scattering (SAXS), the thickness of the crystalline lamellae (mainly containing the double-helices) is 4–6 nm, and the thickness of the amorphous lamellae (containing the main parts of the branch linkages and amylose) is 3–6 nm. ^13^C NMR analysis indicates that much of the amylopectin in the lamellae is in a double-helical arrangement [21]. As visible in light and electron microscopy, the starch granule has alternating 0.1–1 μm thick layers or “growth rings” of different crystallinity. At the level of whole granules, i.e., 1–100 μm, the growth rings are deposited in a mainly concentric manner [22]. However, the relationship between the different structural levels is still obscure, and the outer shapes/morphologies of starch granules cannot yet be deduced from the underlying structural levels. That is, starch molecules packed in the structures on multiple scales have varying degrees of compactness and, thus, show structural heterogeneity. Such heterogeneity has been observed both among different starch granules within the same plant and within the same starch granule.

### 2.2. Blocklet Structures

Very little is known about the starch granular surface structures, which precludes our understanding at the molecular level of the interactions between the degrading enzymes and the granules. However, an interesting layer of structures appears at the organizational level between the lamellar and growth ring structures, namely the so-called ellipsoidal blocklets [23,24]. As described above, a blocklet comprises several semi-crystalline lamellae (Figure 1c). Blocklets can be observed by Atomic Force Microscopy (AFM) and Scanning Electron Microscopy (SEM) as protruding nodules of different sizes (10–500 nm in diameter) on the surface of starch granules [25,26,27,28]. Their size varies between species, ranging from 40 to 100 nm in wheat [27], 10 to 300 nm in potato [29,30], 130 to 250 nm in pea [31], and 10 to 30 nm in corn starch granules [25]. An important hypothesis is that differently structured blocklets form the growth rings so that the more amorphous growth rings consist of smaller and/or less ordered blocklets, while the more crystalline granular layers are formed by larger and/or more perfectly packed blocklets [23]. Even though the blocklet hypothesis remains to be validated, the observed structures can be decisive in delineating the variations in the surface characteristics of starch granules from different plants, i.e., the surface boundary where biosynthesis occurs and hydrolytic enzymes attack.

### 2.3. Starch Granular Topography and Morphology

The inner structure of starch is highly evolutionarily conserved. However, it is still unclear how this leads to the vastly different starch granule morphologies and sizes found in various plant genotypes and organs [32]. For example, among the storage starches, very small starch granules, 0.3–2 μm, are found in quinoa, amaranth and cow cockle [33,34]. Oat, rice and buckwheat have 2–10 μm starch granules [32,35]. Medium-sized (5–30 μm) include granules from cassava, barley, corn and sorghum and large starch granules (up to 100 μm) are found in tubers such as potatoes and canna [36,37]. Bimodal size distributions are characteristic of temperate cereals such as wheat [38] and barley [39]. In mutant plants, especially where the amylose content (AC) exceeds 50%, a significant variation in granular morphology, such as elongated, hollow and aggregated granules, is observed (Figure 2C) [40,41]. It has been suggested that the heterogeneous starch granules within high amylose starches exist in different structures, leading to different properties. For example, aggregate and elongated granules in high amylose maize starches were found to have higher AC and greater thermal and hydrolysis resistance compared to normally shaped granules in these genotypes [40,42]. The size, morphology and number of starch granules can also differ greatly within a single species across different organs and tissues. For example, potato leaf starch granules are smaller than 5 μm, while tuber starch granules range from 10 to 110 μm [43,44].

Notably, many, but not all, types of starch granules have pores on their surface, extending as channels that reach an internal cavity at the granular hilum [45]. Such pores can be observed on the entire surface of corn, sorghum and millet starch granules or along the equatorial groove of wheat, rye and barley starch granules [46]. No porous structures are detected on starch granules of tapioca, rice, oat, canna and arrowroot. Generally, pores were thought to be unique features of starches with A-type crystallinity [47], causing the channels to penetrate into the center of the granules and forming hollow cavities (Figure 2A,B) [48], which are natural features formed during the granules’ development around radially oriented microtubules in the amyloplast [49]. However, the results from mutant and transgenic rice starches have also shown the presence of pores on the B-type crystallinity starch surface [50]. Pores are usually randomly distributed and vary from starch to starch in terms of location, dimensions and extent [47,51,52]. For instance, pores in maize and rice starch granules display diameters in the range of 100 to 200 nm, while pore diameters in wheat are 2 to 3 nm [53]. While channels are native features of granules, their composition and biological significance remain largely unknown [49].

The organization of amylose and amylopectin within starch granules also remains poorly understood. Previous studies have proposed that, for high amylose maize starches, amylose is more concentrated at the periphery than the core, while the opposite is observed for normal and waxy starches [54]. However, such AM distributions can be complex depending on genotypes and mutations in the genes coding for target starch metabolizing enzymes [55,56,57]. Such knowledge, such as the degree of branching and chain length distribution on the surface, is crucial for starch hydrolysis, as the flexible glucan chains on the granule surface are considered a primary factor influencing the susceptibility of starch to amylolysis [58].

Proteins constitute about 0.1–0.8% (*w*/*w*) in cereal, tuber and legume starch granules and are not removed during normal starch fractionation, such as wet milling [59]. Moreover, lipids and proteins may impair the degradation of the starch granule both directly by reducing contact between digestive enzymes and starch and indirectly through reduced swelling of the starch granule during gelatinization as well as gelatinization temperature [60,61,62], even though a typical pure starch granule only contains a small amount of protein (0.1–0.8%) and lipid (0.1–0.7%). All such factors (multi-level starch structures and non-starch compounds) are collectively important for digestibility [63], as discussed further below.

## 3. Action of Hydrolase on Granular Substrates

### 3.1. Hydrolase Mechanisms on Granular Substrates

Enzymatic hydrolysis of starch granules is a complex and heterogeneous catalytic process [64] that depends on the interplay among the widely different granule surfaces and matrix structures described above, the substrate recognition and catalysis by the hydrolase. Importantly, at the granular starch surface, a fraction of glucan chains may be accessible, which potentially provides efficient binding sites and substrates for starch-active enzymes [56,57]. The rate of hydrolysis can be controlled by three main factors: (i) diffusion of the enzyme toward the granule surface; (ii) adsorption onto structures at the granule surface; and (iii) catalytic glycoside bond hydrolysis (Figure 3). Specific interactions between a given enzyme and sites on the granule surface lead to the formation of diverse enzyme–starch complexes (Figure 3(A2,B2)) in which substrate chains adopting productive binding configurations are hydrolyzed, followed by product release. Binding and dissociation at the granular surface are considered favorable for amylases that contain either carbohydrate-binding modules (CBMs), referred to as starch binding domains (SBDs) [65], or so-called surface binding sites (SBSs) [66,67,68] or both, substantially increasing the affinity to granular starch. The lower affinity to the starch model β-cyclodextrin (*K*_d_ = 0.38 mM) of the SBD of CBM family 20 (CBM20) as determined for phosphoglucan, water dikinase, compared to a fungal glucoamylase (*K*_d_ = 0.0075 mM) [69,70] suggested the involvement of dynamic interactions comprising several binding, dissociation and re-binding events. Thus, SBDs assist adsorption at the surface (Figure 3(A2,B2), red frames). Some SBD-containing enzymes may dissociate rather fast from the starch surface (Figure 3(B3), green frames), while others remain bound via the SBD (Figure 3(B3), blue frames) [71]. Although this situation, which illuminates the so-called Sabatier principle, has been demonstrated for cellulose hydrolases [10], a similar analysis for granular starch is only emerging [72]. Accordingly, there are two possible situations during heterogeneous catalysis, namely, desorption-limited catalysis and adsorption-limited catalysis. In the adsorption-limited situation, stronger binding between the enzyme and substrate leads to higher activity: compare Figure 3(A2,B2). However, for desorption-limited catalysis, higher affinity will lead to lower activity due to a too-strong binding of the enzyme [73].

#### 3.1.1. Diffusion Mechanism on the Granular Starch Surface

Diffusion is crucial in interfacial biocatalysis, and the maximum reaction rate depends on a balance between the rate and extent of adsorption and the rate of surface diffusion. Adsorption-limited reactions occur due to an insufficient enzyme population, while high surface (e.g., granular starch) concentrations reduce surface diffusion and enzyme encounter with catalytic sites. This reduction in surface diffusion results from hindered lateral mobility and increased electrostatic interaction strength between the enzyme and substrate [74].

Enzyme surface diffusion is hindered as surface concentration increases, which has been well-documented. For instance, for the digestion of granular starch, starch granules with a smaller size show a larger surface area, which can become a hindrance to enzyme diffusion. Potato starch’s diameter of ~60 μm creates a greater hindrance to enzyme diffusion than the small granule fraction of wheat starch, which has a diameter of ~10 μm and higher binding affinity [75]. Another factor contributing to hindrance in enzyme diffusion is electrostatic interaction, which could arise from starch binding domains and surface binding sites of the enzymes, as explained in Section 3.1.2.

#### 3.1.2. Starch Binding Ability

##### Starch Binding Domains

Among the 97 different CBM families in the CAZy database (http://www.cazy.org/, accessed on 20 December 2022) [76], SBDs constitute CBM20, 21, 25, 26, 34, 41, 45, 48, 53, 58, 68, 69, 74, 82 and 83 [77] and furnish many starch-active enzymes with the enhanced starch binding ability [78]. Structurally, SBDs are recognized as individual immunoglobulin-like fold domains [77,79] of about 100 amino acid residues, except for CBM74, which has about 300 amino acid residues [80]. Functionally, many SBDs are expected to increase the enzyme affinity for granular starch and thereby promote its degradation [77]. SBDs are also hypothesized to bind onto and disentangle interacting α-glucan chains on the starch granule surface, facilitating enzyme hydrolysis [81]. SBDs indeed recognize structures on the granular surface, bringing the catalytic domain (CD) into close contact with the substrate [82], including guiding its α-glucan chains to the active site [81,83].

##### Surface Binding Sites

Surface binding sites (SBSs) are typically identified in crystal structures and then characterized by their aid in site-directed mutagenesis and various biochemical analyses. Functionally, SBSs play essential roles in a range of carbohydrate-active enzymes [84,85,86]. As opposed to SBDs, SBSs are integral to the structure of the CD, appearing as a binding site containing one or two aromatic carbohydrate binding residues, mostly tryptophans and tyrosines [87]. Thus, SBSs can be found on CDs of amylolytic enzymes at a certain distance from the active site or on other domains intimately associated with the CDs [85]. For example, the SBSs in barley α-amylase 1 (AMY1) confer a *K*_d_ of 0.64 mg/mL on barley starch granules [68]. SBSs have been identified in several different enzymes involved in starch degradation, such as α-amylases [85], glucoamylase [88] and pullulanase [89].

#### 3.1.3. General Mechanism of Granular Starch Hydrolysis

In vitro digestion tests have been developed for the dietary degradation of starch utilizing two different enzymes to account for and simulate salivary, pancreatic and small intestinal starch degradation, namely, porcine pancreatic α-amylase (PPA) and glucoamylase (GA) (also named amyloglucosidase (AMG)) from *Aspergillus niger* [90]. In some cases, either PPA [91] or AMG [92] have been used alone. From a mechanistic point of view, all α-amylases share the same type of active site cleft, containing two aspartic acids (Asp) and one glutamic acid (Glu) as catalytic residues. One of the Asp residues is the catalytic nucleophile, Glu is the proton donor and the second Asp residue is a transition state stabilizer [93]. The catalytic mechanism differs according to whether the anomeric configuration of the substrate is retained or inverted in the product. Inverting catalysis involves a single displacement and retaining catalysis involves a double displacement mechanism via the formation and hydrolysis of a covalent intermediate [94]. Enzymes from glycoside hydrolase (GH) families are classified as inverting or retaining enzymes, and while α-amylases and related enzymes are retaining, glucoamylases and β-amylases are inverting enzymes [95].

### 3.2. Dietary Starch Granule Digestion

Starch can provide a very important component in the diet, and most consumed human starchy food is cooked and, thereby, gelatinized. However, most of the starch granule structure is retained in several low-moisture foods, such as muesli, biscuits and some fruits and vegetables [96]. In addition, animal feed, which also attracts attention, contains mainly raw, unprocessed starch, and the importance of such structures has been demonstrated in feed systems [97]. Native starch granule digestion is of interest in light of the increasing recognition of its nutritional advantages due to its higher resistance than gelatinized starch to human and animal amylolytic enzymes and conversion by gut microbiota to compounds beneficial to the host, such as short-chain fatty acids. Understanding the factors that determine the extent of enzyme degradation of granular starches is essential for developing starch-based food and feed with controlled digestibility [2].

#### 3.2.1. Starch Hydrolysis in the Digestive System

Starch digestion in human GIT is a complex biochemical process. It can be divided into oral, duodenal and small intestinal phases, leading to the release of maltooligosaccharides and eventually glucose (Figure 4). 

A classification scheme based on rapidly digestible starch (RDS), slowly digestible starch (SDS) and resistant starch (RS) has been established for nutritional purposes. RDS and SDS are defined as the portions of total starch that are hydrolyzed after incubation with an excess of pancreatic amylase and AMG at 37 °C for 20 min and an additional 100 min, respectively [99]. The non-hydrolyzed residue remaining after 120 min of incubation is classified as RS. However, it has been argued that the 20-min cutoff point used to differentiate between RDS and SDS is too simplistic and lacks physiological significance [99]. RS, on the other hand, is a physiological concept that refers to the portion of starch that reaches the large intestine rather than a precise physical entity [99]. Once RS is transferred to the colon, it can be metabolized by the colonic microbiota [100,101].

RS is metabolized by the human gut microbiota (HGM), providing a range of health benefits, e.g., via the production of short-chain fatty acids [102]. Microbiotal amylases in the colon have been demonstrated to act potently on RS and generate fermentation products, notably butyrate [101], that serve as energy for colonocytes and strengthen the gut barrier, but also other short-chain fatty acids [6]. While mammalian amylolytic enzymes are well characterized, the HGM hydrolases and their action on RS remain understudied. However, RS hydrolytic activity was found in *Ruminococcus bromii* [103], and notably compelling RS-degrading activity was identified in *Bifidobacterium adolescentis* [104]. Hence, efficient degradation of RS by HGM enzymes forms the foundation of RS health-promoting effects.

RS granules constitute an important principle for enhancing the nutritional value of starch through microbial fermentation. High amylose starch (AC > 50%), due to its thermal stability and amylolytic resistance, is considered a good natural source of RS [105]. Such resistance is considered to stem from the dynamic competition between digestion and granular re-organization [72,106]. Still, the mechanisms, product structures and functionalities remain to be validated for different starch types. Notably, the structural re-organization significantly complicates investigations into the interfacial enzymatic hydrolysis of starch.

#### 3.2.2. Glycoside Hydrolases Involved in Human Starch Digestion

Several different hydrolytic activities participate in starch digestion in human GIT. The GH families 13, 14, 15, 31, 57, 119 and 126 are associated with starch degradation [107]. GH13 is among the highly abundant enzyme families within gut bacteria and is most often associated with the initial bacterial processing of starch [108]. Here, we mainly focus on the enzymes applied for starch digestion using in vitro testing protocols.

##### α-Amylase

α-amylases (EC 3.2.1.1) are endo-acting enzymes catalyzing the hydrolysis of internal α-1,4-linkages in starch to mainly generate maltose and maltooligosaccharides [109]. The majority belong to the glycoside hydrolase family 13 (GH13) [110]. They are found in most organisms, including bacteria [111], archaea [112], fungi [113], plants [114] and animals [115,116,117,118]. Notably, α-1,6-linkages in α-glucan chains of AP and glycogen are not hydrolyzed by α-amylases, which release maltose, maltotriose [119], a small amount of glucose [120] and branched α-limit dextrins as final products from starch substrates. In humans, the initial degradation of starch is catalyzed by the human salivary α-amylase (HSA) [115], which varies from person to person because of genetic variation [121]. As a consequence, individuals with higher HSA levels show faster digestion of starch compared with individuals with lower HSA levels [122]. Hence, following HPA-catalyzed hydrolysis, starch is sometimes not fully degraded, depending on the starch structure. According to the definition of RS, i.e., ingested starch reaching the colon, the main step to quantify RS in vitro is to quantify the remaining starch after the removal of the material digested by α-amylase and glucosidase. PPA is mostly used to mimic starch’s upper gut digestion [36,123,124]. The use of PPA to replace HPA for in vitro tests is reasonable since the two enzymes show very high amino acid sequence similarity, including the conservation of surface binding sites 1 and 2 and three catalytic site residues (Asp197, Glu233 and Asp300) and between their three-dimensional structure [116,125] (Figure 5).

Three binding sites for maltooligosaccharides are seen in the crystal structures of PPA and HPA. The two SBSs located at a distance from the active site [126,128] are here named SBS1 (Figure 5B) and SBS2 (Figure 5C). SBS1 in HPA, containing two aromatic residues (Tyr276 and Trp284), and SBS2 (Trp388) are responsible for the granular starch binding (Figure 5B), which has also been identified in barley α-amylase [68] and pancreatic α-amylase [127]. In addition, Ragunath et al. found that the aromatic residue multiple mutants in HSA (W134A/W203A/Y276A/W284A/W316A/W388A) exhibited an 87% reduced ability to hydrolyze granular starch compared with the HSA wild-type [129].

##### Glucoamylase

As mentioned above, four brush border disaccharidase activities, found as two heterodimeric enzymes (MGAM and SI), belong to GH31 [130,131] and are involved in the degradation of starch-derived maltooligosaccharides in the small intestine [132]. These enzymes can act on oligosaccharides produced by α-amylases or directly on starch granules [133]. A widely adopted setup is to use a fungal glucoamylase (GA), also called amyloglucosidase (AMG), to simulate these disaccharidases in combination with porcine pancreatic α-amylase (PPA) to mimic in vitro starch digestion, which is found to give results in line with the findings from ileostomy studies after a fixed time of digestion [134,135]. GA (AMG) (EC 3.2.1.3) catalyzes the hydrolysis of α-1,4- and α-1,6-linkages, releasing β-D-glucose from the non-reducing ends of starch and oligosaccharides. It belongs to GH15 [136] and is found mostly in filamentous fungi [137,138] and in some bacteria [139]. Its specific activity on α-1,6- is only 0.2% of that on the α-1,4-linkages [140].

A C-terminal SBD of CBM20 is found in GA from *Aspergillus niger* [137] and an N-terminal SBD of CBM21 in GA from *Rhizopus oryzae* [141]. Because *Aspergillus niger* GA is typically used to mimic the process of in vitro digestion of starch, only the CBM20 from *A. niger* glucoamylase will be discussed here.

Two binding sites are observed in the three-dimensional structure (PDB ID: 1ACZ) of CBM20 from *A. niger* GA (SBD_GA_) complexed with β-CD [83]. These binding sites differ with respect to function and structure. Binding site 1 contains two tryptophan residues (Trp543, Trp590) and is small and rigid, with an easily accessible planar aromatic face binding to carbohydrates. By contrast, binding site 2 has two tyrosine residues (Tyr527, Tyr556) and is longer and more flexible, possibly guiding the substrate chain to the active site of the CD [83]. Apart from these conserved tryptophans and tyrosines in the two binding sites, there are additional conserved residues that can assist the binding process. A hydrogen bond is present between Lys 578 and maltose in binding site 1. Both binding sites are important for granular starch binding [70,83].

Apart from α-amylase and glucoamylase, there is a series of enzymes that are involved in starch degradation both in nature and in industry, such as isoamylase (EC 3.2.1.68, GH13 and 176) [142], β-amylase (EC 3.2.1.2, GH14) [143,144], α-glucosidase (EC 3.2.1.20, GH13, 31, 76, 97 and 122) [133], pullulanase (EC 3.2.1.41, GH13 and 57) [145] and cyclodextrin glycosyltransferase (CGTase, EC 2.4.1.19, GH13) [146]. The different substrate specificities of these enzymes on a schematic amylopectin segment are shown in Figure 6.

### 3.3. Starch Granule Structure and Digestion

Susceptibility to amylases is seemingly very dependent on the structure of the granular surface, as described above (Section 3.1) [75]. A number of starch granular μm- and nm-scale features have been connected to the susceptibility to amylolytic enzymes. Obviously, amylolytic activity is related to the starch granules’ surface area. Hence, the smaller granules (larger surface area per unit weight) should provide a better substrate than larger ones for both binding and catalysis [147]. However, it should be noted that surface catalysis is a complex process, which is not only determined by granular size [99].

Irrespective of which structural representations that are applied as models for starch granules, e.g., the blocklet and the backbone models, the double-helical chain arrangement forms the foundation of the crystallinity, and despite their tight packing, both these and the amorphous parts of the granule are targets for amylolytic cleavage. However, the number of binding sites available in double-helical structures is limited due to the densely-packed crystalline matrix providing protection for glucosidic linkages from access by enzymes [148]. This effect is evident from the substrate contact where the interaction of a stretch of at least four exposed glucose units is needed between the active site of PPA, which has five substrate binding subsites, and starch [148]. Generally, the less crystalline parts of the starch granules are more readily degraded by amylases than their crystalline counterparts. However, these observations do not apply to high-AM starches, which typically exhibit lower crystallinity than normal starches while also displaying high enzyme resistance [149]. Moreover, the two main crystalline polymorphs found in starch granules show different resistance to amylolysis; the B-type typically being more resistant to enzymatic hydrolysis than the A-type crystalline polymorph for non-mutant crop genotypes [150,151]. However, such resistance is suggested to be more likely due to the distribution of B-type crystallites within the granules and their influence on the local granule organization, such as larger “blocklets” than A-type, rather than the presence of B-type crystallinity per se [99,150] Hence, the crystalline type or amount of crystallinity itself cannot fully explain the enzyme-resistant properties of granular starch. It is likely that the dense packing of the starch chains, hindering enzyme accessibility or catalytic action, is the key factor in determining enzyme resistance, regardless of whether it occurs in the crystal region or the amorphous region [152].

The flexibility of the protruding chains on the surface of starch granules and their chain length are considered crucial factors determining the susceptibility of starch to amylolysis [58]. As shown for pure amylopectin (waxy) rice starch, flexible polysaccharide chains protrude from the granular surface, which are hydrated and highly mobile. Such chains are readily attacked by hydrolytic enzymes and increase with gelatinization [58].

It has been suggested that the hydrolysis of starch granules is primarily dictated by the supramolecular structural (crystalline, blocket and granular surface) characteristics of the granule rather than internal features such as AC and relative crystallinity [153]. These supramolecular structures vary among different types of starch and also within starch granules from the same botanical origin. For example, PPA binds rapidly to a surface with highly disordered α-glucan chains (supposedly no visible blocklet structure), while binding to granules with ordered crystalline surfaces (supposedly large blocklets) is slower (the observed binding rate constant (*k*_obs_) values are 22 × 10^−3^ for pea mutant starch and 2.0 × 10^−3^ s^−1^ for potato) [154]. Moreover, it has been clearly shown that some regions on the periphery surface of the granule are much more resistant to different α-amylases than others, which have been identified as blocklets [155]. Hence, structural heterogeneity within a starch granule population, preferential binding of enzymes to specific regions of a single granule [156], as well as different hydrolysis resistance for starches with different morphologies within high amylose starches exists, as discussed above [40]. However, more knowledge is needed on the surface characteristics, such as blocklets and flexible/amorphous surface patches and chain length distributions, and how they influence enzyme recognition and amylolytic reaction.

An additional layer of complexity is added by taking into account the proteins associated with starch granules, as these can prevent amylolytic attack. Thus, the removal of such bound protein (typically amounting to 0.1–0.8%) enhances the in vitro digestibility of starch (the hydrolysis rate coefficient (*k*, min^−1^) 2- to 3-fold) [62,157,158], demonstrating their physical protective role against amylolytic enzymes [158]. Lipids present another important non-starch component in the starch granule, influencing the starch digestibility by forming starch–lipid complexes, thereby reducing the contact between enzyme and substrate [159]. In conclusion, the µm-scale features of the starch granule and its interaction with lipids and proteins, as well as the nature of the amylase, are decisive for its hydrolytic susceptibility.

## 4. Kinetics of Heterogeneous Systems

### 4.1. Analyzing the Digestive Rate and Efficiency of Granular Starch Degradation

Several attempts have been made to quantify reaction rates of granular starch digestion, especially for α-amylase [117,160,161]. Most protocols include measurements of the extent of hydrolysis over relatively long periods with data fitted as first-order rates of hydrolysis to the Michaelis–Menten (M-M) kinetics model, the latter requiring data for initial rates of hydrolysis. For in vitro digestion of starch or starch-containing foods using α-amylase in combination with AMG over a long period (normally > 120 min), the rate of the reaction typically decays logarithmically with time and hence, the data plot of the concentration of product formed (or quantity of starch digested) against time is linear when plotted on a logarithmic scale (LOS). This substrate decay process fits the standard first-order equation (Equation (1)):(1)$$C_{t}{=C}_{\infty }\left(1\;-{\;e}^{-\mathrm{kt}}\right)$$
where *t* is the digestion time (min), *C_t_* is the fraction of digested starch at digestion time *t*, and *k* is the digestion rate constant (min^−1^). The value of *k* can be obtained from a logarithm of the slope (LOS) plot to avoid using the imprecise value of maximum digestion extent, *C*_∞_ (Equation (2)) [162].
(2)$${\mathrm{In}(dC}_{t}{}_{\;}/dt)=-{\mathrm{kt}+\mathrm{In}(C}_{\infty }k)\;$$

This LOS analysis has been shown to provide reliable linear fits to digestion data obtained from a range of different starch types [153]. The primary substrate structural factors that may affect starch digestion include the particle size (hence, the exposed surface area) of the granule, the presence of pores/crevices in the granule surface, and the supramolecular structure of the carbohydrate chains exposed on the surface of the granule, especially the relative proportions of amorphous and ordered α-glucan chains (the bulk crystallinity of the starch granule and the surface ordering extent of starch), as discussed above [75]. Several studies have demonstrated relationships between starch granule architecture (e.g., granule size, pore size and damaged starch content) and the parameters derived from first-order kinetics, including the digestion rate (*k*) and maximum digestion extent (*C*_∞_) (Equation (1)).

The dimensions of native starch granules are critical with respect to controlling digestion by α-amylase. In general, small starch granules are degraded faster (higher *k* value) than large granules. Compared with the densities of crystalline and amorphous lamellar regions, the presence of granular pores and channels and length of amylopectin “spacers arms” (the link between the double-helices and the amorphous “backbone” of clusters) and branches are more likely to affect enzymatic susceptibility [2]. However, not all the pores facilitate access to a given hydrolase to the inner granule, and such accessibility needs a sufficiently wide pore size (0.4–0.5 μm^2^), while small pores (0.05–0.3 μm^2^) limit granule hydrolysis [163]. In addition, it has been found that the activity at the initial time point of the binding of the enzyme onto the starch granules, as well as surface features (e.g., degree of starch damage), are the primary determinants for the digestion rate and extent (*k* and *C*_∞_ value) rather than the molecular and crystalline structure [164]. However, it has been demonstrated that in rice starch granules, the crystalline polymorph plays a critical role, with B- and C-type being more resistant than A-type due to the higher content of longer chains (DP > 14) present in the B- and C-polymorphic mutant and transgenic rice starch granules compared to A-type granules [50].

However, the *k* value is normally obtained over a long time-course of degradation, and the surface structure of the starch is likely notably changed during this period, including generating and widening of channels, pores and etching of blocklets. Additionally, high amylose starches are prone to molecularly reorganize during digestion [106]. Hence, to determine kinetic parameters for initial catalytic events at the granular surface, assuming that the granular surface is not significantly altered during hydrolysis, forms an important principle for providing more precise information about amylase–starch granule interaction and catalytic effectiveness.

### 4.2. Michaelis–Menten (M-M) Kinetics of Starch with Hydrolase

M-M kinetics provides a relationship between initial, linear reaction rates and substrate concentration. The fundamental form of the M-M kinetics is as follows:*v* = *V*_max_*S*_0_/(*K_M_* + *S*_0_)(3)
*k*_cat_ = *V*_max_/*E*_0_(4)
CE = *k*_cat_/*K_M_*(5)
where *v* is the initial enzyme reaction rate, *K_M_* is the M-M constant, i.e., the substrate concentration yielding an initial rate of *V*_max_/2, as a guide to the substrate availability for a given hydrolase [165]. *V*_max_ is the maximum rate of the reaction, S_0_ is the initial substrate concentration, *k*_cat_ is the catalytic constant representing the turnover number of the enzyme and *k*_cat_/*K_M_* is the apparent second-order rate constant describing catalytic efficiency (CE) [166]. For native granular starches, the experimentally determined apparent *K_M_* is relatively large since only a few of all α-1,4-linkages present in the total starch chains are available for amylase binding and subsequent reaction. Hydrothermal treatment increases the amount of highly accessible starch (gelatinized) and causes *K_M_* to decrease significantly [166].

Several studies have demonstrated that the application of the M-M kinetic model is useful for studying starch granule amylolysis in vitro. Several relationships have been established between the kinetic parameters (*K_M_*, *k*_cat_ and CE = *k*_cat_/*K_M_*) and starch granule characteristics, including surface area and gelatinization enthalpies [165]. The surface area of granules and the degree of order of the starch have an important effect on both the CE and binding rates for amylase to the starch granules. Starch granules with larger diameters (smaller surface area per unit weight) are poorer substrates for amylase, as shown by lower catalytic efficiency, the initial rate of hydrolysis, binding rates and higher *K_M_* values [75,165,167,168]. However, the extent of granule swelling related to the surface area showed a positive relationship with the affinity between starch granules and granular starch hydrolyzing enzyme [169]. In addition, as analyzed by differential scanning calorimetry (DSC) and Fourier transform infrared (FTIR) spectroscopy, a given hydrolase preferentially binds to and catalyzes the degradation of starches having a larger fraction of amorphous material [75,165]. In addition to the structural features of starch granules, other factors, such as the presence of retrograded starch [170] and other polysaccharides, such as cellulose [171], can inhibit the activity of amylase, resulting in a decrease in CE. However, the underlying mechanism is still unknown. In addition, surface barriers, such as surface protein, are also considered as an innate physical barrier against starch–amylase binding, and such protein can even exert inhibitory effects on the porcine pancreatic α-amylase [172]. Wang et al. [173,174] suggested that the access, i.e., binding of enzymes to starch (mainly influenced by the degree of starch gelatinization and porosity), rather than the catalytic hydrolysis, is the rate-determining step for starch digestion. However, supporting experimental data are limited.

Classical M-M kinetics must generally be applied with caution for the amylolytic hydrolysis of granular starch, which is classified as interfacial catalysis. It is characterized by being a two-phase system with a heterogeneous interface, and the reaction steps include diffusion of enzymes to the solid surface, adsorption/binding of enzymes and catalysis of glucosidic bond hydrolysis [152]. A number of concerns have been raised about the practice of applying the conventional or adapted M-M approaches to analyze initial rates for heterogeneous systems, such as that of cellulases acting on insoluble cellulose [11]. The fundamental requirement for the quasi-steady-state assumption (QSSA) of the M-M approach is that the substrate is in large excess with respect to the enzyme. However, for a heterogeneous system, this is difficult to fulfill experimentally since the true molar concentration of the substrate cannot be defined unambiguously. As an effect, it can be argued if the traditional M-M approach can be applied at all to interfacial reactions between starch granules and amylases.

An additional complication is related to the lack of knowledge on in vivo ranges of amylase and starch. It has been suggested that the concentration of human α-amylases is in the range of 5 to 15 nM [166]. However, HPA is also suggested to be present at very high concentrations in the lumen of the small intestine—10–65 mg/mL starch. With a specific activity of 1000–9000 units/mL amylase, 1 unit of amylase activity catalyzing the formation of 1 mg of maltose/h amounts to approximately a 10^5^ times higher activity level than the 5–15 nM maltose/h mentioned above [175]. Typically, also in living cells, enzyme concentrations are comparable to those of the substrate, which means that a considerable fraction of the available substrate can be bound to the enzyme [176]. Hence, in many biologically relevant situations, enzyme kinetics in vivo operate under conditions where the substrate is not in vast excess, particularly for starch granules that have a limited attackable site. Therefore, the application of traditional M-M kinetics for starch granule degradation can lead to inaccurate estimates of *K_M_* and *V*_max_ parameters due to the dynamic substrate concentration and uncertain quasi-steady-state assumption.

Notably, the brush border disaccharidases (α-glucosidases) in the intestinal villus can have relevance to granular starch digestion in vivo (Section 3.1). These enzymes primarily hydrolyze maltooligosaccharides derived from starch α-amylolytic action. However, it has been suggested that these enzymes not only passively convert the end products of α-amylase digestion to absorbable glucose but are capable of acting directly on granular and polymeric starch [133]. Hence, analysis of the kinetics for brush border α-glucosidases or the model enzyme AMG used in an in vitro digestion test is also physiologically relevant. Following the small intestinal degradation of starch and uptake of glucose, degradation of RS in the colon relies on hydrolysis by a group of gut microbes’ amylases [177] and further fermentation into beneficial metabolites such as butyrate. While the mammalian hydrolases are very well characterized, gut microbiotal starch hydrolases are less well studied. RS hydrolytic activities are identified in so-called amylosomes in *Ruminococcus bromii* [103], and recently, strong RS-degrading activity was found in *Bifidobacterium adolescentis* [178]. The efficient degradation of RS by these microbial enzyme systems provides the foundation for the health-promoting effects of RS along with other dietary fibers. Contrary to what is observed for the effect of AM content on the action of PPA relevant to gastrointestinal digestion [99], the influence of the AM content on the fermentation rates in the large intestine was small. Thus, when wheat starches with different AM contents (37–93%) were tested, similar fermentation kinetics and amylolytic enzyme activities (microbial α-amylase, β-amylase, pullulanase and glucosidase) were found, irrespective of the AM content [179]. However, more types of RS substrate need to be analyzed, including partly degraded granular starch.

### 4.3. Handling Interfacial M-M Kinetics at High Amylase Concentrations

Since the reaction between digestive enzymes and the starch granule occurs in a two-phase system, the reaction mechanism involves a kinetically-significant adsorption step as well as the actual catalysis, as discussed above. It has been suggested that hydrolysis resistance in starch has two fundamental origins: binding-limiting, where binding to the enzyme is the limiting factor; or hydrolysis-limiting after binding, where the hydrolysis step itself is limiting after binding has occurred. Therefore, it is important to determine which limitation, or combination, is responsible for resistance in a particular starch or food [99].

Bright-field and fluorescence microscopy have enabled the observation of the binding of active and inhibited hydrolases to starch granules, revealing several important findings: (1) the surface structure and botanical origin of the starch granules affect amylase binding, with maize starch granules showing more enzyme binding than potato starch granules; (2) the surface of starch granules is heterogeneous as an effect of digestion, with certain regions being more susceptible to enzymic degradation [156]; and (3) two catalytic patterns have been observed for a given enzyme on different starch types, depending on whether the granules have surface pores (an “inside-out” digestion pattern) or not (an “outside-in” digestion pattern) [156]. Additionally, the two patterns have also been observed for the same starch and different amylases, depending on the SBD engineering of the amylases, i.e., whether there is a flexible linker between the CD and SBD or not [155]. While microscopy is a valuable tool for visualizing the location of enzymes during binding and hydrolysis, it cannot provide quantitative information, such as the number of bound enzymes, binding rates and binding affinity. Kinetic analyses, such as the Freundlich [154] and Langmuir isotherms [72], have been applied for this purpose. Additionally, correlation analysis between affinity and activity, based on the Sabatier principle, has also been used to obtain more information about the binding and hydrolysis mechanisms [10]. However, experimental details must be carefully considered to avoid changes to the starch granule surface during binding analysis with hydrolases, such as reacting under non-hydrolyzing conditions (e.g., low temperature (0 °C) or the addition of an inhibitor).

In classical M-M enzyme kinetics, measuring catalysis for two freely diffusible enzymes and substrate species provides an important experimental link between starch structure and hydrolysis efficiency. However, as already mentioned, it is unclear whether conventional M-M theory, which requires a large excess of substrate, is biologically fully relevant for interfacial starch amylolysis. It was recently demonstrated that using microcrystalline cellulose from wood as the model solid substrate for cellulolytic enzymes, simple steady-state kinetics can be applied to heterogeneous substrate systems [11] by varying the enzyme concentration instead of the substrate concentration (Figure 7) and introducing a factor ^kin^Γ_max_ (in the unit mol/g) which enumerates sites available for enzyme attack per gram of substrate (Equation (6)). This approach requires experiments that include both enzyme and substrate saturation (i.e., conventional and inverse M-M combined). When combined with Langmuir isotherm data for enzyme adsorption to the surface of starch granules, it is possible to calculate the density of attack and binding sites in relation to different surface structures. Such data can provide important insights into the mechanisms of amylolytic reactions for a range of starch substrates, especially related to whether the reaction is limited by binding or catalysis [11,72]. Moreover, starch granule degradation efficiency at high enzyme concentrations in vivo (where all attack sites are complexed with enzyme and free enzyme accumulates in the aqueous phase) could also be compared.
^kin^Γ_max_ = *K_M_*/*K*_½_(6)
where *K*_½_ is the mass load at substrate half-saturation given by the conventional M-M, and *K_M_* is the molar concentration of attack sites that gives enzyme half-saturation given by the inverse M-M.

Notably, the relationship between the rate of glucosidic bond hydrolysis and enzyme concentration is not exactly linear for a granular starch system since not all the enzyme was able to bind to the surface, and not all the binding events are productive, as discussed above. Such effects are important to quantify and need to be addressed. For cellulases, it has been shown that not all binding sites are competent for catalytic conversion [11], and hence, degradation can be limited by either enzyme adsorption, catalysis or both. The method has been validated in great detail as applicable to cellulases (references here), but its application for granular starch systems is in its infancy and just emerging [72]. Such combined kinetic/adsorption data, including the density of attack sites, adsorption capacity and the ratio between them, will shed new light on the rate-determining step of the starch digestion with respect to binding and/or catalytic action (Figure 8), as prospected before [99], and provide an estimate of the overall efficiency for digestive enzyme and granular starch. Moreover, it will provide unprecedented insight into the role of enzyme-accessible surface features manifested for a range of different starch granules. In conclusion, enzyme coverage at the granular surface and the ratios of binding and catalytic events can provide data to explain the fundamental mechanisms underlying RS functionality for different starch types and amylases.

## 5. Conclusions and Prospects

Interfacial enzyme reactions are natural for starch granules, but our understanding of the kinetics remains far less developed than for enzyme reactions in solution. From a nutritional perspective, native starch is an important source of RS, which is physiologically useful for reducing risk factors such as obesity and type 2 diabetes [180]. However, the widely varying susceptibilities of RS towards amylase adsorption and amylolytic attack are only vaguely understood, and major obstacles include a lack of deeper analysis of granular starch surface hydrolysis-related structural characteristics and reliable kinetics approaches [99]. A better understanding of the structural basis for determining starch digestibility could help design starch foods with enhanced nutritional outcomes. It is hoped that more techniques will be developed for analyzing the surface of starch granules, particularly to clarify the structural characteristics of regions that exhibit varying degrees of resistance to enzyme degradation.

An in vitro digestion approach provides an efficient key experimental link between the starch structure and nutritional function in vivo. Instead of only comparing experimental data of starch digestion, the digestion curves can be modeled by kinetic equations (first-order kinetics and M-M kinetics and their derivations (Section 4.3)). Such kinetic analysis helps evaluate the role of starch structural features in determining starch digestibility. We suggest that combined effects of the glucosidic scissile bond stability, weak substrate adsorption, low substrate accessibility or a combination of these are responsible for the starch granular-hydrolysis resistance difference (Figure 8). However, whether in vitro conditions corroborate the assumption of substrate excess and hence, the ubiquitous QSSA is questionable for applying the conventional M-M kinetics to granular starch degradation [11]. The LOS method, defined by a first-order equation, is widely applied and describes long-term dynamic digestion well. However, this approach suffers from not taking into account possible and likely, decreases in enzyme activity during the long incubation course applied and the changing structure of the substrate surfaces. An inverse M-M kinetics model developed for cellulases [11] provides initial rates and adsorption specific for given substrate surfaces and enzyme activities, thereby providing a new manageable way forward to gain insight into the impact of the structure of substrate surfaces available for enzyme interaction and catalysis (Figure 8). We hope the information provided here can support the development of new ways forward, carefully evaluating different kinetic and adsorption methods for granular starch and starch-degrading enzymes to provide reliable relational data on enzyme binding and catalysis and substrate structures for relevant research, e.g., nutritional assets of starch.

## Figures and Tables

**Figure 1 molecules-28-03799-f001:**
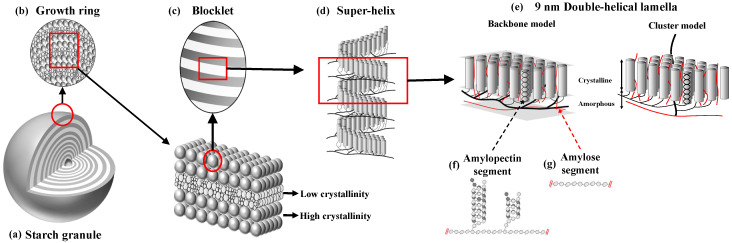
The multi-level structure of the starch granule as depicted by the blocklet [8] and backbone organization [9]. (**a**) Starch granule, (**b**) growth rings as a repeating layered structure with a period of a few hundred nanometers, containing a semi-crystalline region (high crystallinity) and an amorphous region (low crystallinity), (**c**) spherical blocklets with a diameter between 10 and 300 nm in the semi-crystalline regions, (**d**) left-handed amylopectin super-helix consisting of alternating crystalline lamellae (containing the linear parts of the chains) and amorphous lamellae (containing most of the branch points) which stack with a periodicity of ~8–11 nm (**e**), and molecular structure of (**f**) amylose and (**g**) amylopectin.

**Figure 2 molecules-28-03799-f002:**
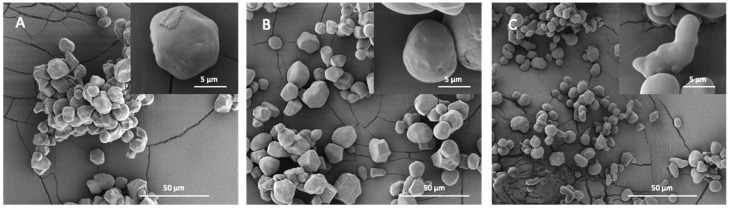
Maize starch granular morphology with different amylose content: 0% (**A**); 27% (**B**); and 72% (**C**).

**Figure 3 molecules-28-03799-f003:**
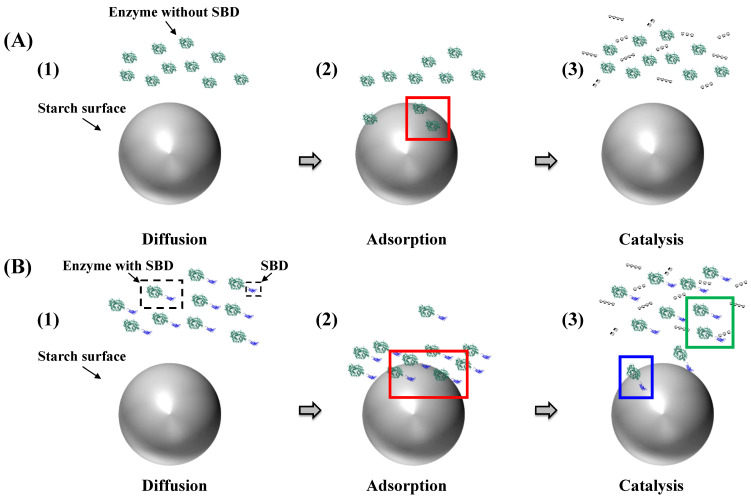
Heterogeneous catalysis by amylases without (**A**) or with (**B**) SBD attacking blocklets on the starch granule surface during diffusion (1), adsorption (2) and catalysis (3). Red frames: Enzyme adsorption. Blue frame: Amylases remaining attached. Green frame: Fast dissociation.

**Figure 4 molecules-28-03799-f004:**
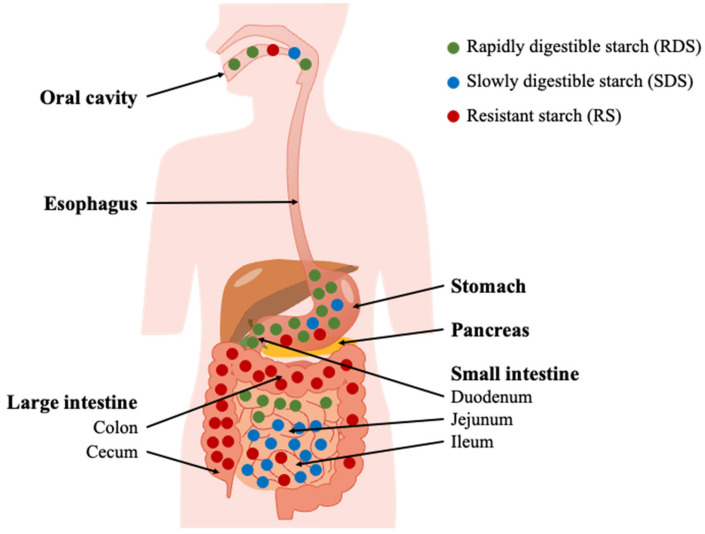
Schematic of starch digestion *in vivo* denoting the fates of RDS, SDS and RS in the GIT (adapted with permission from Ref. [98]. 2023, Andreas Blennow).

**Figure 5 molecules-28-03799-f005:**
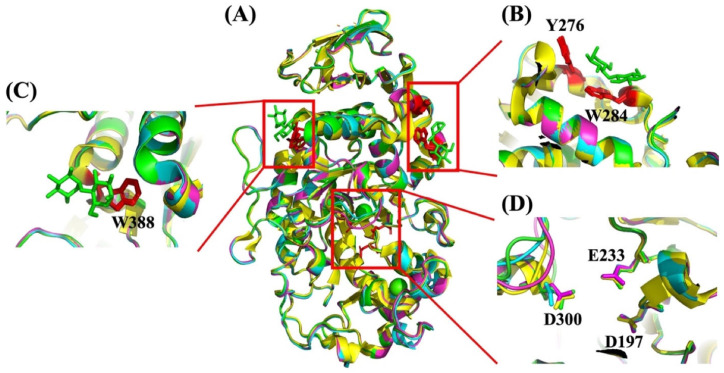
Crystal structure alignment between HPA (PDB: 1HNY (Green) [116]; 3IJ8 (Yellow) [126]) and PPA (1PIF (Blue) [127]; 1PIG (Violet) [127]). (**A**) Full-length structures. (**B**) Surface binding site 1 (SBS1, red) in PPA (PBD: 1PIG) in complex with maltose (Green), (**C**) Surface binding site 2 (SBS2, red) in PPA (PBD: 1PIG) with maltose as a ligand (Green) and (**D**) Active site with catalytic residues in stick representation.

**Figure 6 molecules-28-03799-f006:**
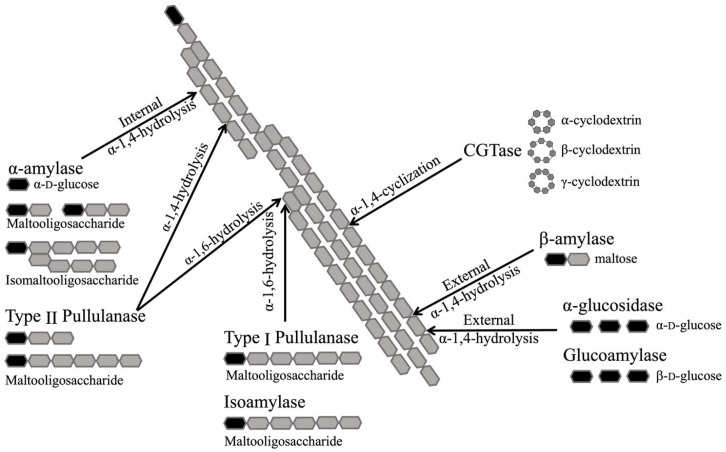
Specificities of different starch hydrolytic enzymes on a schematic amylopectin segment and related products. Grey: glucose unit; black: reducing end glucose unit or glucose molecule.

**Figure 7 molecules-28-03799-f007:**
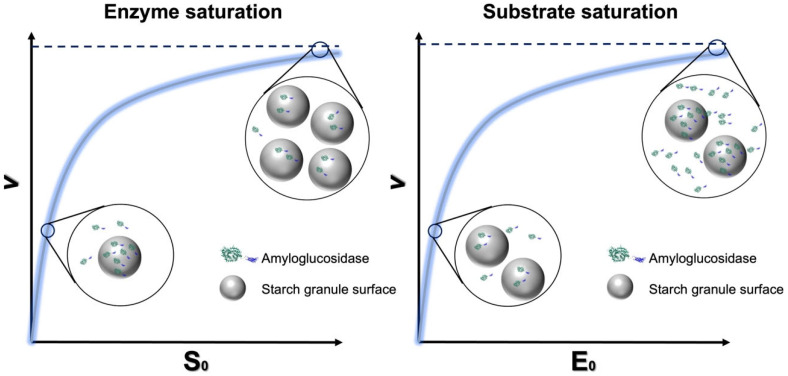
Schematic principle of interfacial M-M kinetics at the excess substrate (**left**, conventional M-M) and excess enzyme (**right**, inverse M-M) (amyloglucosidase as a model enzyme) (adapted with permission from Ref. [11]. 2023, Yu Wang).

**Figure 8 molecules-28-03799-f008:**
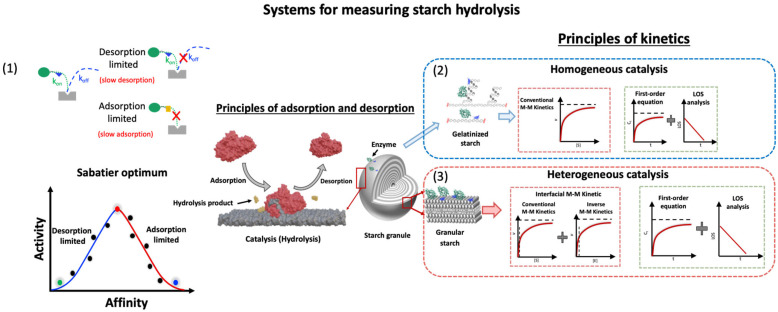
Schematic approaches for analyzing enzyme kinetics for starch-acting hydrolases. Hydrolysis of starch in its gelatinized or granular states is different. Enzymatic hydrolysis of starch granules is a heterogeneous catalytic process with an insoluble substrate acted on by a soluble enzyme, while the hydrolysis of gelatinized starch is a well-defined homogeneous system where both enzyme and substrate are in solution. Granule hydrolysis is controlled by adsorption, catalysis and desorption. (**1**) The Sabatier principle states that optimal catalysis occurs when the catalyst and reactant interact with intermediate affinity. However, this principle has not been used extensively for bioanalytical heterogeneous catalysis yet, such as starch granules. (**2**) Conventional M-M theory, which requires a large excess of substrate, is widely applied in homogeneous catalysis, while it has limited applicability for studying catalysis on an insoluble starch granular substrate. A combined “inverse M-M strategy” provides a description of granular starch interfacial enzyme reactions with readouts revealing the density of enzyme attack sites on the substrate surface as probed by a given enzyme. (**3**) The widely applied LOS analysis provides catalytic rates over extended time frames but does not take into account initial rates. Especially for a granular starch system, the starch surface is likely notably changed due to its long time reaction, generating an ill-defined catalytic process (adapted with permission from Ref. [72]. 2023, Yu Wang).

## Data Availability

All available data are included in the article.

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
