# Peer review of "Interfacial Catalysis during Amylolytic Degradation of Starch Granules: Current Understanding and Kinetic Approaches"

_molecules, 2023, doi:10.3390/molecules28093799_

Round 1

Reviewer 1 Report

This paper reviews “Interfacial catalysis in amylolytic degradation of starch granules – mechanisms and experimental approaches”.

This is an interesting topic and the paper provides a well-written overview of current knowledge and status, including suggestions for methods to be used for analysis.

Title: “amylolytic degradation of starch granules – mechanisms”, mechanisms have not been solved yet? Is it not too early to refer to mechanisms in the title?

Lines 39-41: the Introduction states “Hydrolytic degradation of such granules in the human gastrointestinal tract (GIT) occurs in consecutive oral, duodenal and small intestinal stages”, but does not specify what enzymes are involved. Please clearly separate degradation of cooked starch and granular starch, here, and throughout the manuscript.

A relevant recent review paper of this group is missing under references? Zhong et al. (2022) Recent advances in enzyme biotechnology on modifying gelatinized and granular starch. Trends in Food Science & Technology 123, 343–354.

Line 186: [56,57nis the same as ref 13857], please check.

Lines 238-253: Paragraph “3.1.2. Mechanism of granular starch hydrolysis” deals with current knowledge about degradation of solublized starches. Please specify what is known about degradation of granular starches?

Reference 166: not complete, please check.

Reviewer 2 Report

In the review by Tian et al. the reactions of amylolytic degradation of starch granules is in focus. The topic is of high interest and also timely and in principle I strongly support the publication of this review. A lot was summarized in this review and that was much work. However, I see some weak points, which might be considered by the authors. First, in my impression the focus gets a little lost in the beginning (Introduction, starch granules). A lot of data regarding starch is summarized, but unfortunately mostly superficial. A few examples: starch is not only an energy storage, the size limits are further random, resistant starch is not well defined in the beginning when first used, the connection of amylose and amylopectin ratio to RS is missing, the legend of Figure 1 is insufficient, the separation of both models is missing, the cluster model is insufficient explained, the allomorphs where mentioned but the structural differences are not explained, surface near glucans are not considered at the granule structure… In 2.3 morphology is mentioned in the headline, but no morphology data are given in this paragraph.

Further, the authors emphasise that the granule degradation is important in planta but never refer to transitory starch.

Thus, this part can be shortened or the mentioned points have to be explained in more detail.

The more focused part starting with the action of hydrolases is much better, but also needs to be clarified, thus e.g. what is the background of beta-cyclodextrins as starch model, the headlines are in parts misleading, thus in the mechanism of granule starch hydrolysis nearly no information is given regarding the headline, similar in case of dietary starch granule digestion; what is the background of Figure 4, AlphaFold? Further, there is an inhomogeneity, thus EC numbers were used extensively in some parts and in others not. Overall the reader still get a strong impression that the sections where written by different authors and not polished in the end. In the title experimental approaches were mentioned, but unfortunately it is not included in the manuscript. Overall, a lot was tried to combine but the connection is not always done well.

Further points:

There are a number of small mistakes, thus revision of the text is necessary, similar several references are missing (e.g. line 114-116).

Line 186: the reference is incorrect

Line 237: incorrect references, as in a review not published data should be avoided

Line 288-296: similar problem of unpublished data

Line 379ff: the statement that smaller granules should provide a better substrate is unclear to me and needs further explanation

Line 607: Enzyme kinetics is the key experimental link between the structure and function in biocatalysis in vitro. What is the aim of this fundamental statement? Why it is restricted to in vitro, why key experimental link?..

Round 2

Reviewer 2 Report

The authors have adapted the manuscript and have considered most points raised.